# Factors Associated with Willingness toward Organ Donation in China: A Nationwide Cross-Sectional Analysis Using a Social–Ecological Framework

**DOI:** 10.3390/healthcare11060824

**Published:** 2023-03-10

**Authors:** Mengjun Zeng, Haomiao Li, Xiaohui Song, Jipin Jiang, Yingchun Chen

**Affiliations:** 1Department of Health Management, School of Medicine and Health Management, Tongji Medical College, Huazhong University of Science and Technology, Wuhan 430030, China; 2Key Laboratory of Organ Transplantation, Ministry of Education, NHC Key Laboratory of Organ Transplantation, Institute of Organ Transplantation, Tongji Hospital, Tongji Medical College, Huazhong University of Science and Technology, Wuhan 430030, China; 3School of Political Science and Public Administration, Wuhan University, Wuhan 430072, China; 4Department of Cardiology, Renmin Hospital, Wuhan University, Wuhan 430060, China; 5School of Basic Medical Science, Wuhan University, Wuhan 430062, China; 6Research Centre for Rural Health Service, Key Research Institute of Humanities & Social Sciences of Hubei Provincial Department of Education, Wuhan 430030, China

**Keywords:** organ donation, willingness, social–ecological, quantile regression, China

## Abstract

Improving public willingness toward organ donation is an important solution to the low organ donation rate. This study aimed to explore factors impacting public willingness for organ donation in China from a multi-agent perspective and further explore the impact of these factors on high or low willingness, using a social–ecological framework. Data from a total of 11,028 (effective rate, 94.18%) participants were analysed. Generalised linear model (GLM) and quantile regression were used to explore factors associated with willingness and high/low willingness toward organ donation, respectively. The mean willingness toward organ donation was 56.9 (range, 0–100) points. GLM regression revealed that age, family health, males, lower educational levels, and agricultural hukou were negatively associated with willingness. For personality, conscientiousness was negatively associated with willingness, whereas openness was positively associated with willingness. Health literacy perceived social support, and media utilisation were positively associated with willingness. Quantile regression further indicated that educational levels of college, bachelor, master’s, and PhD, openness, health literacy, perceived social support, and media utilisation were positively associated with organ donation willingness at all percentiles. It is necessary to adopt more targeted and diversified publicity, education, and guidance for different types of individuals. Meanwhile, social support needs to be strengthened. To enhance the willingness of the residents to donate organs, media publicity should be strengthened, particularly by using modern ways to improve their health literacy.

## 1. Introduction

Organ transplantation plays an essential role in the treatment of patients with organ failure. With advances in medical technology, organ donation has saved large numbers of lives. In China, organ donation and transplantation have shown remarkable improvement through the untiring efforts of several generations of transplant surgeons [1]. The team of transplant coordinators is growing and maturing with the construction and improvement of the organ donation system. Voluntary deceased organ donation has increased annually since the pilot programme was initiated in 2010. Since 1 January 2015, China abolished the use of prison organs, and voluntary organ donation has become the only legal source of organ transplantation in China [2]. At present, China still adopts the ‘opt-in’ system, which means people with full capacity for civil conduct can express their willingness to donate through voluntary registration, and people under the age of 18 are forbidden to register. Volunteers can choose the type of organ or human tissue they want to donate. When the intention to donate changes, the registration can be changed or withdrawn at any time. However, the ultimate donation is subject to medical evaluation and the consent of the immediate family. There are two main approaches to registering as a volunteer, which, respectively, are online registration through WeChat or the official website of the China Organ Donation Administrative Center and, offline, through written registration through the local Red Cross. If a citizen has not expressed his disapproval of organ donation during his lifetime, his immediate family members may jointly express their willingness to donate his organs in written form [1,3]. From 2015 to 2020, 29,334 cases of organ donation after death were completed. The organ donation rate (per million population (PMP)) increased from 2.01% in 2015 to 3.70% in 2020. Although the absolute number of organ donations in China has increased, like other countries in the world, China still faces a shortage of donors to meet the huge demand for domestic patients [4]. The PMP in China, in 2020 (3.70%), was lower than the global average (5.80%), and much lower than other countries, such as the United States (38.0%), Spain (38.0%), Estonia (25.4%), Croatia (25.4%), and Portugal (24.8%) [5].

The Chinese government has made great efforts to improve public awareness and willingness for organ donation. The main publicity channels for organ donation include shooting public service advertisements. To be more specific, celebrities who register as organ donation volunteers are invited to serve as ambassadors for organ donation. In addition, news media report typical donation cases and touching stories to motivate more people to join. Moreover, public welfare activities are held on special festivals—June 11th has been set as China’s Organ Donation Day since 2017. Each year on this day, Red Cross, organ donation management centres, and Organ Procurement Organizations from the national to the provincial level will hold various forms of publicity activities, such as organ donations in communities, universities, hospitals, and so on. Moreover, organ donation courses have been introduced into Chinese university classrooms. In the relevant departments of the hospital, such as the ICU and neurosurgery department, organ donation billboards will be posted in order to let the family members of severe patients know that organ donation could be another option when a life cannot be saved [1,6]. At present, the public’s willingness and awareness of organ donations are increasing annually; however, the organ donation rate still lags far behind the advanced international level.

Improving public willingness toward organ donation is a significant strategy for improving the low organ donation rate. Therefore, identifying factors associated with willingness toward organ donation is essential and urgent. Previous studies have explored factors associated with willingness toward organ donation from many perspectives, mainly focusing on the following aspects. First, demographic factors, including age, gender, occupation, income, and family-centred traditional values, have been reported to be associated with willingness toward organ donation [7,8]. Second, knowledge of organ donation; attitude toward organ donation and factors influencing this attitude, such as information delivery, education, and media use, have been reported to be associated with willingness toward organ donation [9,10,11,12]. Third, laws, legislations, and social policies, such as incentives, are also associated with willingness toward organ donation [13]. Text messaging, feedback, and prosocial emotions also affect willingness toward organ donation [14]. In addition, some personality-associated characteristics, such as self-efficacy, may be linked with willingness toward organ donation. Self-efficacy is a cognitive process in which individuals learn new behaviours that affect their ability to their performance in future events through environmental and social factors [15]. Some previous studies have indicated that personality, determined using the Big Five personality traits, influences willingness toward blood and organ donation [16,17,18], nevertheless, their associations with willingness toward organ donation are less studied.

Previous studies have revealed that the factors associated with willingness toward organ donation belong to various dimensions and originate from multiple subjects, such as individual, family, society, and government. Nevertheless, only a few studies have assessed factors associated with willingness toward organ donation from a multi-agent perspective. Meanwhile, to improve willingness, factors need to be intervenable, although most factors identified in previous studies (such as age, gender, marital status, etc.) are difficult to intervene. In addition, one potential factor may have different effects on high and low willingness, respectively; considering these different effects is important for constructing more targeted intervention measures.

The present study was based on national survey data and can provide more representative conclusions and make more general suggestions. On one hand, the present study comprehensively analyses the factors influencing willingness toward organ donation from a multi-agent perspective. On the other hand, it focuses on the differences in the factors affecting the different levels of willingness. We used the social–ecological model as a framework to assess factors influencing willingness toward organ donation. The social–ecological model is a multiple-tier framework, which includes individual, relational, community, and societal levels from micro to macro levels, for organising risk and protective factors, which then aid in determining corresponding prevention strategies. In a previous study, the social–ecological model proposed that individual willingness was associated with individual, physical, social, and regulatory influences [19]. Efforts to modify the willingness must consider these multiple levels of constraints [20,21,22]. Factors toward organ donation willingness identified in current research tend to be fragmented. That is, even where summaries of factors are provided, they are often limited to one or two social–ecological levels. Following the enhanced organisation of factors, the social–ecological model of organ donation willingness can provide grounding for multi-level intervention and prevention programme design and implementation [23].

In summary, based on a nationwide survey, this study aimed to explore factors influencing public willingness toward organ donation and further explore the impact of these factors on high or low willingness using a social–ecological framework. To add to the current literature on the topic and further comprehend organ donation willingness, as well as intervention feasibility of factors, our study particularly detects several elements in each level or in several levels that are significantly associated with organ willingness.

## 2. Materials and Methods

### 2.1. Sampling and Participants

The data used were from surveys conducted in 23 provinces, 5 autonomous regions, and 4 municipalities in mainland China from July to September 2021. A total of 120 cities were selected from 2 to 6 cities in each province and autonomous region by random number table method in multi-stage sampling. Based on the results of the seventh National Population Census in 2021, 120 urban residents were selected for quota sampling (quota attributes are gender, age, and urban–rural distribution) so that the gender, age, and urban–rural distribution of the samples basically conform to the population characteristics. In each sample city, questionnaires were collected through a professional online questionnaire survey platform (Wenjuanxing: https://www.wjx.cn/; accessed on 1st July to 1st September 2021). The volunteers were asked to distribute the questionnaires face-to-face with a quick response code, which can present the questionnaires once scanned. Informed consent was obtained from all respondents before the survey began. A total of 11,709 questionnaires were collected. The datasets are not publicly available as the data needs to be used for other research purposes but are available from the corresponding author upon reasonable request.

Participants who were aged ≥12 years and filled in the informed consent form, could complete the network questionnaire survey by themselves or with the help of the investigators and could understand the meaning of each item in the questionnaire were included in the present study. After excluding participants with key missing values, 11,028 valid questionnaires (with a 94.18% effective rate) were obtained, with high quality and national representativeness.

### 2.2. Variables

#### 2.2.1. Dependent Variable

Participants were asked to choose a number from 0 to 100, representing the willingness toward organ donation, with 100 being the strongest level of willingness, and 0 indicating no willingness at all. Residents chose scores according to their own intentions to reflect their levels of willingness.

#### 2.2.2. Independent Variables

Based on a social–ecological framework, we classified factors that may potentially be associated with willingness toward organ donation into the following three levels: individual, family, and social levels (Figure 1).

**Individual level**. The individual level refers to the demographic and behavioural factors that influence willingness toward organ donation. At the individual level, participants’ demographic indicators (age, gender, educational level, work status, and nationality) were selected. Additionally, self-efficacy, personality and health literacy were taken into analysis. Self-efficacy was measured using the New General Self-Efficacy Scale (Appendix A) [24]. Furthermore, personality is closely related to distress, anxiety, and several behaviours [25]. Personality was measured using the 10 items of the Big Five Inventories, with the following domains: neuroticism, extraversion, openness to experience, agreeableness, and conscientiousness (Appendix A) [26]. Health literacy includes a set of skills required to make appropriate health-related decisions [27], which was associated with attitude, knowledge, and behaviours and can enable individuals to develop transferable skills in accessing, understanding, analysing, and applying health information [28,29]. Health literacy (Appendix A) was measured using the New Short Form Health Literacy Instrument [30].

**Family level**. At the family level, factors included income, marital status, family health, hukou status, and family type. The family health was measured using the Short Form of the Family Health Scale (FHS-SF) translated into Chinese with the consent of the original author (Appendix A) [31]. Family health is a resource at the level of the family unit that develops from the intersection of the health of each family member, their interactions, and capacities, as well as the family’s physical, social, emotional, economic, and medical resources. The Chinese version of the FHS-SF has good reliability (Cronbach alpha = 0.83) and validity (χ^2^/df = 4.28, GFI = 0.98, NFI = 0.97, RFI = 0.95, RMSEA = 0.07 < 0.08) and can be used to assess the level of family health of Chinese residents [31,32]. Hukou indicates the respondent’s hukou place (including non-agricultural hukou and agricultural hukou) and is a special identifier in China. The hukou status affects several aspects of life in China, including buying a house, buying a car, children’s school enrolment, and other welfare [33,34]. Family type includes nuclear family, conjugal family, backbone family, single-parent family, and other types.

**Social level**. At the social level, factors included social support and social media utilisation. Social support was measured using the Perceived Social Support Scale (Appendix A). Perceived social support is defined as the availability of individuals to make one feel cared about, valued, and loved [35]. Social media included newspapers, magazines, radios, televisions, books (not textbooks), computers (including tablets), and smartphones. Participants were asked about the frequency of using these media (0 = never, 1 = occasional, 2 = sometimes, 3 = often, and 4 = almost every day). Subsequently, social media utilisation was calculated as the sum of all the frequencies (Appendix A).

### 2.3. Statistical Analysis

First, for description analysis, we compared the differences in willingness between different socioeconomic groups using the Kruskal–Wallis one-way analysis. Kernel density estimations were performed to display the distribution of willingness toward organ donation, as well as the willingness among different groups. Smooth curve fitting for the trend of willingness along with the change of continuous variables (self-efficacy, personalities, family health, health literacy, perceived social support, and media utilisation) were performed on the basis of generalised additive or linear models.

Second, to explore the influences of all factors on willingness toward organ donation, a generalised linear model (GLM) was applied. Although independent variables were classified into different levels, our data did not comply with the hierarchical data, in which sample households were included in each sample region, and each individual was included in each sample household [36]. Therefore, multi-level analysis is not appropriate for this study. We constructed three GLM models, i.e., a model including only individual characteristics, a model including individual characteristics and family characteristics, and a model including individual, family, and social characteristics. The fitness of the models was measured based on log-likelihood, Akaike information criterion (AIC) and Bayesian information criterion (BIC).

Third, to explore the different effect size predictions of impacting factors, quantile regression (QR) was utilised. QR, introduced by Koenker and Bassett (1978), is an extension of classical least squares estimation of conditional mean models to the estimation of an ensemble of models for several conditional quantile functions. QR can describe the relationships between the explanatory variables and willingness across the entire distribution by enabling the modelling of any conditional quantile of the outcome variable [37]. In addition, QR does not assume the normality or homoscedasticity of the distribution of outcome variables [38]. Quantiles 25, 50, and 75 were analysed. 

The *p* values were two-tailed, where statistical significance was set at an alpha level of 0.05. Data were analysed using Stata 17.0.

### 2.4. Ethics Approval and Consent to Participate

This quantitative study was performed in accordance with the ministry of health and ‘involves people of biomedical research ethics review method (try out)’, national drug supervision and administration of the quality control standard for clinical trials (2003), medical instrument clinical trial regulations (2004), and Declaration of Helsinki. The investigators obtained ethics approval from the Ethics Committee of Jinan University (JNUKY-2021–018). We certify that all applicable institutional and governmental regulations concerning the ethical use of human volunteers were followed over the course of this study. All the study participants provided written informed consent to participate in this study upon recruitment.

## 3. Results

Of the 11,028 participants, the mean score of organ donation willingness was 56.9 (range, 0–100) points. The distribution of organ donation willingness indicated that the distribution of each group was relatively balanced, with the highest willingness accounting for the largest and the lowest willingness accounting for the second largest, presented in Figure 2. The sample characteristics and organ donation willingness across different groups are shown in Table 1. Almost 60% of the respondents were aged 19–45 years old; 54.37% were females; 57.66% were registered as non-agricultural hukou; 56.44% were married; 52.13% of respondents had an education level of college or bachelor. The distribution of the sample complied with the distribution of the national population. Younger participants (under 45 years old), females, higher income, those with non-agricultural hukou, higher educational levels, students, and those with free medical care had higher willingness toward organ donation, whereas widowed participants had the lowest willingness (37.41 ± 33.62). The highest willingness was exhibited among those with educational levels of master’s and PhD (65.91 ± 30.96), followed by students (64.01 ± 31.01).

The distribution of organ donation willingness across different groups is presented in Figure 3, which provides further information about the relatively high and low willingness, particularly the high density of high willingness in groups of non-agricultural hukou, high educational levels, and students, and the high density of low willingness in groups of older adults, widowed participants, and agricultural hukou.

The factors associated with willingness toward organ donation based on GLM are shown in Table 2. We sequentially included individual characteristics (Model 1), family-level factors (Model 2), and social-level factors (Model 3) in the models, with model 3 revealing the best fitness, with the lowest AIC and BIC. In model 3, age (β = −1.65, 95% confidence interval [CI] = −2.77 to −0.53), and family health (β = −0.19, 95% CI = −0.32 to −0.06) were negatively associated with willingness. Females (β = 2.08, 95% CI = 0.89–3.28) and students (β = 4.21, 95% CI = 2.02–6.41) had higher willingness than males, unmarried and currently occupied participants. Participants who had educational levels of college and bachelor’s (β = 5.64, 95% CI = 2.04–9.23) and master’s and PhD (β = 9.98, 95% CI = 5.72–14.24) had higher willingness than those who were illiterate. For personality, conscientiousness (β = −0.84, 95% CI = −1.29 to −0.40) was negatively associated with willingness, whereas openness (β = 1.04, 95% CI = 0.61–1.47) was positively associated with willingness. Participants with agricultural hukou (β = −2.28, 95% CI = −3.59 to −0.96) were negatively associated with willingness compared to those with non-agricultural hukou. Health literacy (β = 0.31, 95% CI = 0.19–0.44), perceived social support (β = 0.12, 95% CI = 0.06–0.19), and media utilisation (β = 0.59, 95% CI = 0.46–0.72) were also positively associated with willingness.

The results of QR, which estimated the different effect sizes of each factor on different willingness segments, are revealed in Table 3. It was intriguing that for educational levels of college and bachelor and master’s and PhD, the personality of openness, health literacy, and media utilisation were positively associated with organ donation willingness at all percentiles. Student occupation was positively associated with willingness only at the 50th percentile. The personality of conscientiousness and agricultural hukou were negatively associated with willingness at the 25th and 50th percentiles. Females, secondary education and college, and a bachelor’s were associated with higher willingness at the 50th and 75th percentiles. Social support was associated with higher willingness at the 25th and 50th percentiles.

## 4. Discussion

In the present study, factors associated with a willingness toward organ donation in China were comprehensively identified and described using a social–ecological framework, which is derived from individuals, families, communities, and the whole society. Such an analytical framework provides a good reference for improving national organ donation willingness in the future, i.e., systematic efforts need to be made from multiple levels and multiple dimensions. In addition, this study provides a more systematic idea for studying organ donation willingness and identified more factors that could be manipulated through interventions compared with previous studies.

In this study, age, male sex, lower educational levels, and agricultural hukou were negatively associated with a willingness toward organ donation. These results are consistent with those of several previous studies [39,40]. Older populations and males are not that willing to donate organs due to traditional family values [41]. Influenced by traditional concepts, Chinese people believe that keeping the body intact shows their respect for the deceased. In addition, owing to the lack of death education, Chinese people are very taboo on the topic of ‘death’, so few people will talk about what they will do after death while they are alive. Urban residents are proposed in wider and more abundant information than rural residents, which could lead to a gap in knowledge about organ donation between urban and rural residents [41]. Similarly, the willingness to donate organs develops with an increase in education level; individuals with higher education levels are more likely to attend organ donation education programmes than those with lower education levels [42]. Even if these factors are challenging to intervene with, we can adopt differentiated guidance and publicity methods to improve the willingness toward organ donation in the future. In addition, the subsequent focus of organ donation publicity should be placed on death education, which means introducing the concept of organ donation into death education. Furthermore, integrating the idea of organ donation into hospice care is of great significance in nudging the penetration of organ donation in China.

Personality is associated with donating willingness and behaviour. Previous studies indicated that the donors frequently make a very conscious choice consonant with their personality, ranging from autonomous, nonconformist, headstrong, and self-determined to a prosocial attitude. In other words, the donors make a symbolic statement following their self-identity [16]. It is intriguing that different dimensions of personality play different roles in our study, which indicated that openness was associated with higher willingness, but conscientiousness was negatively correlated with willingness. Openness describes the individual acceptance of new things and multiple orientations. In China, organ donation is often irrelevant to most people’s daily lives. Therefore, the more open to new things, the more organ donation can be accepted. Nevertheless, in the cultural background of China, it is important to keep the body intact and not be disfigured [43]. Some residents may think that organ donation, which makes their bodies not intact, may be disagreed with by their family members and, thus, be considered a sign of irresponsibility to their families. Therefore, conscientiousness is negatively related to organ donation. This can also explain why family health, which is closely related to family members’ conscientiousness, was negatively correlated with willingness. The higher the level of family health, the higher the degree of dependence and trust among family members, that is, the greater the influence of family on individual willingness.

Health literacy is associated with knowledge and attitude toward donation, including how and why to donate, as well as the significance of donation. This has also been supported by previous studies [44,45]. Additionally, in China, health literacy affects the ability to make autonomous decisions related to health and may reduce the influence of traditional beliefs [46]. Therefore, improving public health literacy in several ways may be effective in improving the willingness toward organ donation.

The positive association between social support and willingness toward organ donation could be explained by the following viewpoints. First, social support is associated with rapid information diffusion and can improve organ donation awareness [47,48]. Second, social support could solve the concerns about organ donation. Previous studies have indicated that ambivalence is common among donor candidates; however, instrumental social support can mediate the negative effects of donation-related concerns. Recommendations include providing appropriate social support to minimise donation-related concerns, thereby reducing the ambivalence of donation candidates [49,50].

Social media has been proven to be an important publicity channel to improve the public willingness toward organ donation, which is also supported by previous studies [51,52]. Particularly, with the popularisation of information technology, network media have been integrated into every aspect of life and became significant sources of information. This should also be an important means to publicise the social significance of organ donation and related processes and policies.

### Limitations

Although this study used a nationally representative database, it had several limitations. First, this study suffered from the inherent flaws of a cross-sectional study, and causal effects could not be obtained. Second, we conducted the survey through respondents retrospectively completing a questionnaire, which may be subjected to recall bias. Third, the survey mainly concentrated on the health indicators of Chinese populations. Therefore, factors associated with organ donation willingness may not be well-rounded. More factors should be explored in future studies. Fourth, we applied a social–ecological framework to identify factors from a more systematic perspective. However, our data were not multi-level; therefore, they could not perfectly support the application of the socio–ecological model in the field of organ donation. Fifth, the study participants were asked to indicate their willingness toward organ donation on a scale from 0 to 100, which may be an overly fine-grained scale, even though we conducted quantile regressions, which may overcome this limitation to a certain extent. In light of these limitations, subsequent prospective studies are needed to examine the most effective measures to improve the public willingness toward organ donation.

## 5. Conclusions

In China, residents’ willingness toward organ donation needs to be improved. Improving the willingness requires taking measures from multiple levels, such as the individual, families, and society. More targeted and diversified publicity, education, and guidance for different types of individuals should be adopted. Meanwhile, social support needs to be strengthened. To enhance the residents’ willingness to donate organs, media publicity should be strengthened, particularly by using modern ways, to improve their health literacy.

## Figures and Tables

**Figure 1 healthcare-11-00824-f001:**
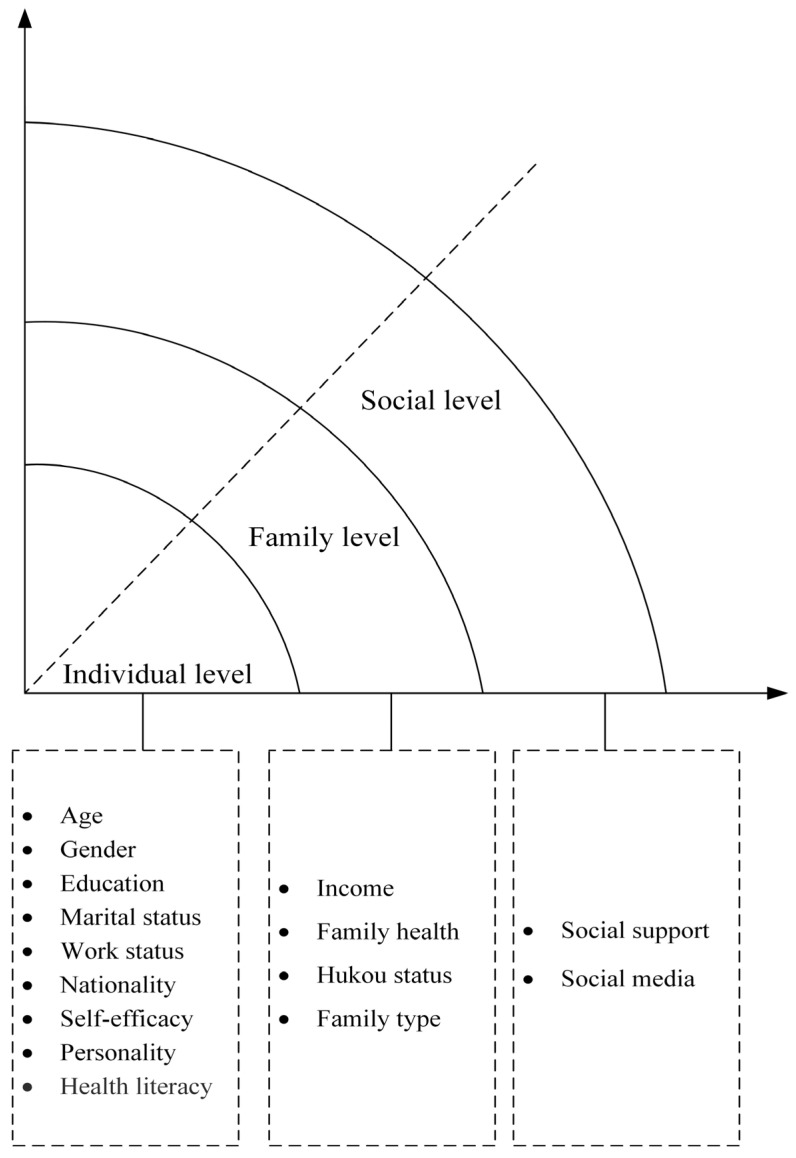
Conceptional model of social-ecological framework.

**Figure 2 healthcare-11-00824-f002:**
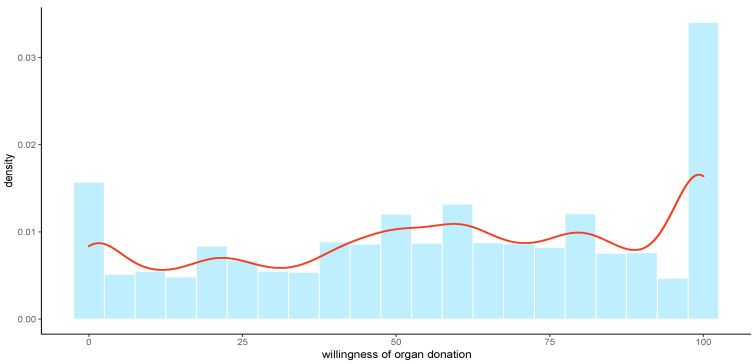
The distribution of organ donation willingness through Kernel Density Estimation.

**Figure 3 healthcare-11-00824-f003:**
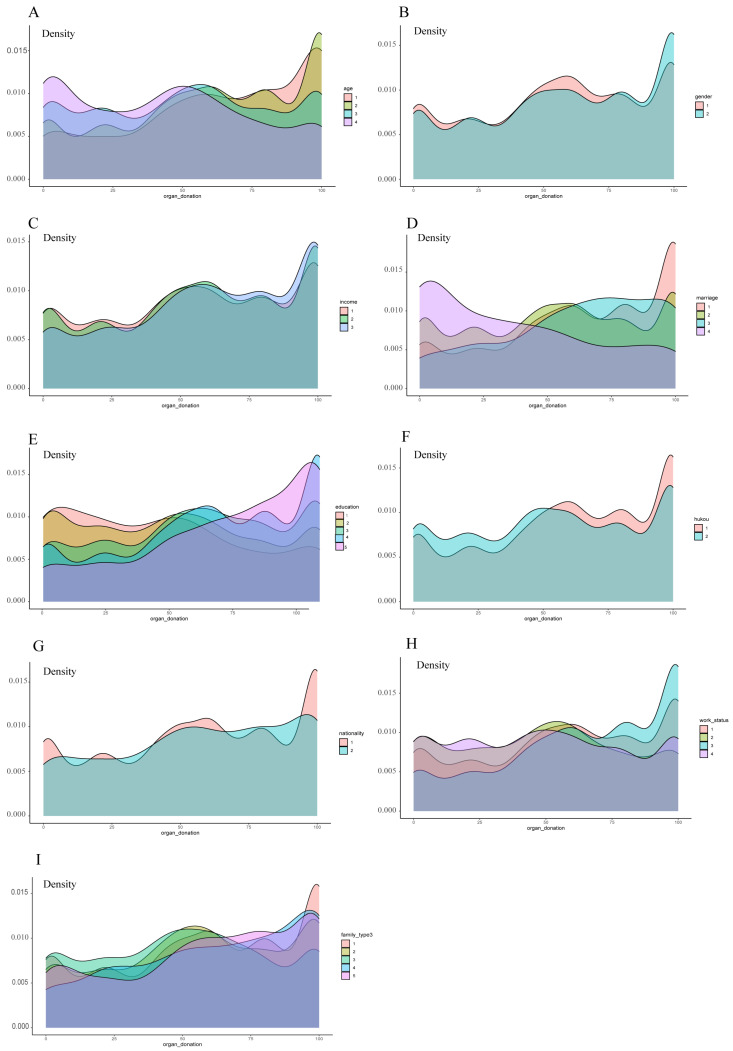
The distribution of organ donation willingness among different groups through Kernel Density Estimation. The groups are: (**A**) age (1 = under 18 y; 2 = 19–45 y; 3 = 46–65 y; 4 = above 65 y); (**B**) gender (1 = male; 2 = female); (**C**) income (1 = “≤3000”; 2 = “>3000 and ≤7500”; 3 = “>7500”); (**D**) marital status (1 = unmarried; 2 = married; 3 = divorced; 4 = widowed); (**E**) education level (1 = Illiteracy; 2 = below secondary school; 3 = secondary education; 4 = bachelor degree; 5 = master’s degree or above); (**F**) hukou (1 = non-agriculture; 2 = agriculture); (**G**) nationality (1 = han nationality; 2 = ethnic minorities); (**H**) work status (1 = working; 2 = retired; 3 = student; 4 = unfixed occupation); (**I**) family type (1 = nuclear family; 2 = conjugal family; 3 = stem family; 4 = single-parent family; 5 = other).

**Table 1 healthcare-11-00824-t001:** Participants’ characteristics and willingness of organ donation.

	N (%)	Willingness	*p* Value
Age			<0.001
≤18 y	1065 (9.65%)	62.59 ± 32.02	
19–45 y	6601 (59.84%)	59.70 ± 31.72	
46–65 y	2490 (22.57%)	51.63 ± 32.35	
>65 y	875 (7.93%)	44.25 ± 32.37	
Gender			<0.001
Male	5033 (45.63%)	55.54 ± 31.98	
Female	5998 (54.37%)	58.10 ± 32.62	
Monthly per capita household income, RMB			<0.001
≤3000	3246 (29.43%)	55.35 ± 32.61	
>3000 and ≤7500	5325 (48.27%)	56.61 ± 32.37	
>7500	2460 (22.30%)	59.72 ± 31.82	
Hukou status			<0.001
Non-agricultural hukou	6360 (57.66%)	59.11 ± 32.01	
Agricultural hukou	4671 (42.34%)	53.96 ± 32.59	
Educational level			<0.001
Illiteracy	378 (3.43%)	43.34 ± 32.90	
Below secondary school	2188 (19.84%)	47.66 ± 32.45	
Secondary Education	1978 (17.93%)	55.81 ± 31.76	
College and Bachelor	5750 (52.13%)	60.59 ± 31.60	
Master and PhD	737 (6.68%)	65.91 ± 30.96	
Work status			<0.001
Working	4637 (42.04%)	57.05 ± 32.20	
Retired	884 (8.01%)	48.55 ± 31.98	
Student	3314 (30.04%)	64.01 ± 31.01	
No fixed occupation	2196 (19.91%)	49.38 ± 32.36	
Marital status			<0.001
Unmarried	4363 (39.55%)	62.87 ± 31.57	
Married	6226 (56.44%)	53.36 ± 32.10	
Divorced	207 (1.88%)	61.55 ± 30.79	
Widowed	235 (2.13%)	37.41 ± 33.62	
Family type			<0.001
Nuclear family	6547 (59.35%)	57.80 ± 32.50	
Conjugal family	1763 (15.98%)	56.04 ± 31.80	
Backbone family	1345 (12.19%)	50.86 ± 31.59	
Single-parent family	418 (3.79%)	61.31 ± 32.15	
Other	958 (8.68%)	59.28 ± 32.54	
Nationality			0.745
Han nationality	10,386 (94.15%)	56.91 ± 32.36	
Ethnic minorities	645 (5.85%)	57.33 ± 32.30	

**Table 2 healthcare-11-00824-t002:** Factors associated with willingness of organ donation based on generalized linear regressions.

Factors	Model 1		Model 2		Model 3	
	Coefficient	Confidential Level	Coefficient	Confidential Level	Coefficient	Confidential Level
**Individual level**						
Age	−2.198 ***	−3.289 to −1.107	−1.678 **	−2.805 to −0.551	−1.65 **	−2.77 to −0.53
Gender (Ref. Male)	1.842 **	0.647 to 3.036	2.051 **	0.854 to 3.249	2.08 **	0.89 to 3.28
Educational level (Ref. Illiteracy)						
Below secondary school	0.393	−3.151 to 3.937	0.103	−3.421 to 3.627	−0.57	−4.08 to 2.95
Secondary Education	5.751 **	2.090 to 9.413	4.864 **	1.201 to 8.526	3.42	−0.24 to 7.09
College and Bachelor	8.842 ***	5.280 to 12.405	7.591 ***	4.011 to 11.172	5.64 **	2.04 to 9.23
Master and PhD	14.202 ***	10.016 to 18.388	12.433 ***	8.2 to 16.666	9.98 ***	5.72 to 14.24
Work status (Ref. Working)						
Retired	−0.811	−3.433 to 1.810	−0.936	−3.583 to 1.711	−1.44	−4.07 to 1.19
Student	4.189 ***	2.611 to 5.767	4.606 ***	2.415 to 6.796	4.21 ***	2.02 to 6.41
No fixed occupation	−0.817	−2.646 to 1.012	−0.529	−2.387 to 1.33	−0.27	−2.11 to 1.58
Nationality (Ref. Han nationality)	1.396	−1.098 to 3.890	1.149	−1.358 to 3.656	0.89	−1.61 to 3.39
Self-efficacy	0.174 *	0.039 to 0.309	0.257 ***	0.112 to 0.402	0.02	−0.14 to 0.18
Personality						
Extraversion	−0.215	−0.625 to 0.193	−0.215	−0.625 to 0.195	−0.26	−0.67 to 0.15
Agreeableness	−0.401	−0.853 to 0.051	−0.169	−0.641 to 0.303	−0.24	−0.71 to 0.23
Conscientiousness	−0.968 ***	−1.411 to −0.524	−0.872 ***	−1.318 to −0.426	−0.84 ***	−1.29 to −0.4
Neuroticism	−0.306	−0.756 to 0.143	−0.307	−0.756 to 0.142	−0.28	−0.73 to 0.17
Openness	1.212 ***	0.780 to 1.643	1.136 ***	0.703 to 1.569	1.04 ***	0.61 to 1.47
Health literacy	0.412 ***	0.291 to 0.534	0.423 ***	0.3 to 0.546	0.31 ***	0.19 to 0.44
**Family level**						
Income (Ref. ≤ 3000)						
>3000 and ≤7500	--	--	−1.099	−2.551 to 0.353	−1.18	−2.63 to 0.27
>7500	--	--	−0.766	−2.57 to 1.037	−1.04	−2.84 to 0.76
Marital status (Ref. Unmarried)						
Married	--	--	−0.604	−2.714 to 1.505	−0.86	−2.98 to 1.25
Divorce	--	--	3.106	−1.737 to 7.948	2.62	−2.19 to 7.43
Widowed	--	--	−9.147 ***	−14.119 to −4.176	−8.79 ***	−13.77 to −3.80
Family health	--	--	−0.158 **	−0.276 to −0.04	−0.19 **	−0.32 to −0.06
Hukou status (Ref. Non-agricultural hukou)	--	--	−2.506 ***	−3.824 to −1.188	−2.28 ***	−3.59 to −0.96
Family type (Ref. Nuclear family)						
Conjugal family	--	--	1.089	−0.611 to 2.789	0.86	−0.83 to 2.56
Backbone family	--	--	−0.692	−2.596 to 1.212	−0.79	−2.69 to 1.10
Single-parent family	--	--	4.929 ***	1.565 to 8.292	5.19 **	1.83 to 8.56
Other	--	--	4.66 ***	2.513 to 6.808	4.38 ***	2.24 to 6.52
**Social level**						
Perceived social support	--	--			0.12 ***	0.06 to 0.19
Media utilization	--	--			0.59 ***	0.46 to 0.72
**Log-likelihood**	−53,640.098		−53,604.467		−53,555.300	
**AIC**	9.729		9.724		9.716	
**BIC**	1.07 × 10^7^		1.06 × 10^7^		1.05 × 10^7^	

Note: * *p* < 0.05; ** *p* < 0.01; *** *p* < 0.001. AIC: Akaike information criterion; BIC: Bayesian information criterion.

**Table 3 healthcare-11-00824-t003:** Factors associated with high/low willingness of organ donation based on quantile regression.

Factors	Quantile 25		Quantile 50		Quantile 75	
	β (95%CI)	*p* Value	β (95%CI)	*p* Value	β (95%CI)	*p* Value
**Individual level**						
Age	−2.53 (−4.18, −0.88)	0.003	−1.04 (−2.18, 0.09)	0.071	−2.24 (−3.64, −0.84)	0.002
Gender (Ref. Male)	2.15 (0.16, 4.14)	0.034	2.2 (0.88, 3.52)	0.001	2.12 (0.6, 3.64)	0.006
Educational level (Ref. Illiteracy)						
Below secondary school	−2.83 (−6.54, 0.87)	0.134	1.35 (−3.7, 6.39)	0.6	1.38 (−4.2, 6.96)	0.629
Secondary Education	2.98 (−1.67, 7.64)	0.209	6.55 (2.26, 10.84)	0.003	5.4 (−1.17, 11.98)	0.107
College and Bachelor	5.99 (1.87, 10.11)	0.004	9.44 (5.36, 13.51)	<0.001	7.74 (2.56, 12.93)	0.003
Master and PhD	10.52 (3.65, 17.39)	0.003	15.38 (10.15, 20.61)	<0.001	10.69 (5.04, 16.33)	<0.001
Work status (Ref. Working)						
Retired	−0.98 (−5.24, 3.28)	0.652	−1.83 (−5.18, 1.52)	0.284	−3.27 (−6.27, −0.28)	0.032
Student	5.95 (1.15, 10.76)	0.015	4.76 (1.73, 7.8)	0.002	2.73 (−0.44, 5.90)	0.092
No fixed occupation	−0.06 (−3.36, 3.24)	0.971	−1.64 (−4.55, 1.27)	0.269	−1.69 (−3.98, 0.60)	0.149
Nationality (Ref. Han nationality)	−0.14 (−3.38, 3.10)	0.933	2.35 (−1.60, 6.30)	0.243	0.41 (−2.98, 3.81)	0.811
Self-efficacy	−0.01 (−0.40, 0.37)	0.947	0.05 (−0.18, 0.27)	0.701	0.08 (−0.11, 0.28)	0.394
Extraversion	−0.20 (−1.07, 0.67)	0.653	−0.44 (−1.14, 0.27)	0.226	−0.25 (−0.68, 0.18)	0.255
Agreeableness	−0.36 (−1.35, 0.63)	0.477	−0.52 (−1.28, 0.25)	0.185	0.12 (−0.40, 0.64)	0.658
Conscientiousness	−1.3 (−2.16, −0.44)	0.003	−1.22 (−2.13, −0.31)	0.009	−0.41 (−1.03, 0.20)	0.184
Neuroticism	−0.58 (−1.57, 0.40)	0.247	−0.43 (−1.05, 0.19)	0.17	−0.43 (−1.01, 0.16)	0.154
Openness	1.46 (0.61, 2.31)	0.001	1.88 (1.10, 2.67)	<0.001	0.99 (0.38, 1.61)	0.002
Health literacy	0.29 (0.05, 0.53)	0.018	0.40 (0.23, 0.57)	<0.001	0.65 (0.45, 0.85)	<0.001
**Family level**						
Income (Ref. ≤3000)						
>3000 and ≤7500	−0.44 (−2.98, 2.10)	0.735	−2.23 (−4.12, −0.33)	0.021	−1.05 (−2.8, 0.7)	0.238
>7500	−0.27 (−3.43, 2.88)	0.865	−2.41 (−5.66, 0.85)	0.147	−1.32 (−3.62, 0.97)	0.259
Marital status (Ref. Unmarried)						
Married	−1.57 (−5.5, 2.36)	0.433	−0.72 (−3.84, 2.39)	0.649	−1.99 (−4.89, 0.9)	0.177
Divorced	0.82 (−7.17, 8.8)	0.841	0.01 (−10.6, 10.62)	0.998	1.87 (−2.72, 6.46)	0.425
Widowed	−11.09 (−15.59, −6.6)	<0.001	−14.31 (−24.61, −4.02)	0.006	−7.98 (−18.64, 2.68)	0.142
Family health	−0.47 (−0.71, −0.24)	<0.001	−0.1 (−0.3, 0.11)	0.357	−0.02 (−0.2, 0.16)	0.824
Hukou status (Ref. Non-agricultural)	−2.97 (−4.80, −1.15)	0.001	−3.31 (−5.44, −1.18)	0.002	−1.1 (−2.84, 0.64)	0.216
Family type (Ref. Nuclear family)						
Conjugal family	1.36 (−1.59, 4.31)	0.366	0.62 (−1.2, 2.44)	0.506	−0.15 (−2.91, 2.62)	0.917
Backbone family	2.52 (−0.73, 5.77)	0.129	−0.94 (−3.83, 1.95)	0.525	−3.33 (−6.99, 0.33)	0.074
Single-parent family	6.42 (−2.78, 15.61)	0.171	9.48 (3.59, 15.37)	0.002	5.01 (1.54, 8.48)	0.005
Other	5.87 (2.03, 9.71)	0.003	5.42 (2.11, 8.74)	0.001	3.04 (0.26, 5.81)	0.032
**Social level**						
Perceived social support	0.19 (0.07, 0.30)	0.001	0.19 (0.10, 0.28)	<0.001	0.06 (−0.01, 0.13)	0.119
Media utilization	0.78 (0.57, 0.99)	<0.001	0.73 (0.49, 0.96)	<0.001	0.49 (0.30, 0.68)	<0.001

## Data Availability

The datasets generated and/or analysed during the current study are not publicly available due the data still needs to be used for other research but are available from the corresponding author upon reasonable request.

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
