# Peer review of "Factors Associated with Willingness toward Organ Donation in China: A Nationwide Cross-Sectional Analysis Using a Social–Ecological Framework"

_healthcare, 2023, doi:10.3390/healthcare11060824_

Round 1

Reviewer 1 Report

Review comments of manuscript healthcare-2224393 titled:

Factors associated with willingness toward organ donation in China: a nationwide cross-sectional analysis using a social-ecological framework

In this manuscript, the factors influencing public willingness toward organ donation were explored based on a nationwide survey, and the impact of these factors on high or low willingness was further explored by using a social-ecological framework. The results showed that the mean willingness toward organ donation was 56.9 (range, 0-100) points. The factors of age, family health, males, lower educational levels, and agricultural hukou were negatively associated with the willingness. The factors of openness, health literacy, perceived social support, and media utilization were positively associated with willingness, whereas conscientiousness was negatively associated with willingness. And thus, the results suggested that the residents' desire to donate organs could be enhanced by strengthening media publicity.

I think the manuscript has made an objective and detailed investigation of the factors affecting organ donation, and I would like to address several points as follows:

Minor comments:

1.It is suggested that the general information of participants, such as age, sex, education, place of origin, etc 

2.On page 8, Fig3, the scale identification of the coordinate axis was not clear. 

Author Response

1.It is suggested that the general information of participants, such as age, sex, education, place of origin, etc 

Response: We are grateful for the reviewer's constructive insights to our manuscript, and apologize for the unclear statements. We supplemented these information in the “Results” section. The details are as follows:

Almost 60% of the respondents were aged 19-45 years old; 54.37% were females; 57.66% registered as non-agricultural hukou; 56.44% were married; 52.13% respondents had education level of college and bachelor.

2.On page 8, Fig3, the scale identification of the coordinate axis was not clear. 

Response: We apologize for the unclear scale identification of the figure. We have revised the scale identification of the coordinate axis, which is shown in the main text.

Reviewer 2 Report

Dear authors and editors,

Thank you for the manuscript on this important issue in the health care system! I fully agree that research on factors influencing willingness to organ donation is crucial and I appreciate the approach using a multiple tier framework, which includes quite a few potentially influential factors.

There are three general aspects that I would like to comment on regarding the manuscript:

1.)    I miss a more detailed description of how the organ donation system in China is currently working. That is, how are people informed about organ donation (you mention organ donation education programmes)? Which incentives are used? How do people opt for or against organ donation and are there options to specify, e.g., which organs they (don not) want to donate or to define a person who might decide on that if they are deceased? How are they able to change their decision? Explaining these details would provide a good basis for suggestions of measures to increase the willingness to organ donation in China in your discussion.

2.)    In the media and in the literature, the issue of forced organ harvesting in China is quite present (see, e.g., Analysis of official deceased organ donation data casts doubt on the credibility of China’s organ transplant reform | BMC Medical Ethics | Full Text (biomedcentral.com) , China forcefully harvests organs from detainees, tribunal concludes (nbcnews.com) ). I suggest that the authors include this issue in the introduction and discussion, especially as they refer to the official donation and transplantation rates and refer to them as both “ranking second in the world” (l. 51) and “lag[ging] far behind the advanced international level” (l. 55).

3.)    This is a methodological question. The study participants were asked to indicate their willingness to organ donation on a scale from 0 to 100. To me, that seems to be an overly fine-grained scale for a question that, actually, is answered with “yes” or “no”. After all, someone can either agree to organ donation and be an organ donor or reject.  Please elaborate on the meaning of the numbers. For example, what does it mean if someone indicates their willingness with “50”? Does it mean that this person is indecisive, or that they “do not care”, or that they “somehow agree”? I would like to ask the authors to clarify the meaning of the numbers on this scale and the reasons why they chose this scale (rather than, e.g., a Likert scale).

Some minor comments:

Ll. 66/67: You mention “incentives” twice, here.

L. 78: This sentence is a bit confusing. The authors write about topics of previous studies and then state that “Meanwhile, the present study has some limitations.” , most likely referring to their own study. Usually, the limitations are discussed in the discussion section. The first listed aspect does not seem to present a limitation, however, as it is a new feature of the here presented study. The fact that many of the investigated factors (age, gender, being a widow, etc.) cannot be manipulated through interventions is indeed important and should be mentioned in the discussion again. I suggest rephrasing the whole paragraph.

L. 106/108: “organ willingness” – probably, “donation” is missing?

L. 118: “a professional online questionnaire platform” – please name the platform

L. 139: Please explain: Why is “marital status” rather an individual than a family level?

L. 215: “interviewees” – rather “study participants”?

Ll. 304-316: The argumentation in this paragraph is unclear to me. The fact that organ donation is rare presents a contrast to your former statement that it is “ranking second in the world” (l. 51). I do not understand how the sentence starting with “Therefore” follows from the statement before. Furthermore the authors need to elaborate how “conscientiousness” relates to “the concept of family”, which is brought up in the next sentences, and also why organ donation should be considered a sign of irresponsibility to the family (at least if they are just talking about deceased donation).
